# Disrupted Resting State Attentional Network Connectivity in Adolescent and Young Adult Cannabis Users following Two-Weeks of Monitored Abstinence

**DOI:** 10.3390/brainsci12020287

**Published:** 2022-02-18

**Authors:** Julia C. Harris, Alexander L. Wallace, Alicia M. Thomas, Hailey G. Wirtz, Christine M. Kaiver, Krista M. Lisdahl

**Affiliations:** Psychology Department, University of Wisconsin-Milwaukee, 2441 E. Hartford Ave, Milwaukee, WI 53211, USA; harri585@uwm.edu (J.C.H.); walla228@uwm.edu (A.L.W.); aliciathomasphd@gmail.com (A.M.T.); hgwirtz@uwm.edu (H.G.W.); cmkaiver@uwm.edu (C.M.K.)

**Keywords:** adolescence, cannabis, attentional networks, neuroimaging

## Abstract

Background. Numerous neuropsychological studies have shown that cannabis use during adolescence and young adulthood led to deficits in sustained and selective attention. However, few studies have examined functional connectivity in attentional networks among young cannabis users, nor have characterized relationships with cannabis use patterns following abstinence. Methods. Differences in resting state functional connectivity (RSFC) within the dorsal (DAN) and ventral (VAN) attention networks were examined in 36 adolescent and young adult cannabis users and 39 non-substance using controls following two weeks of monitored abstinence. Observed connectivity differences were then correlated with past-year and lifetime cannabis use, length of abstinence, age of regular use onset, and Cannabis Use Disorder symptoms (CUD). Results. After controlling for alcohol and nicotine use, cannabis users had lower RSFC within the DAN network, specifically between right inferior parietal sulcus and right anterior insula, as well as white matter, relative to controls. This region was associated with more severe cannabis use measures, including increased lifetime cannabis use, shorter length of abstinence, and more severe CUD symptoms. Conclusions. Findings demonstrate that regular cannabis use by adolescents and young adults is associated with subtle differences in resting state connectivity within the DAN, even after two weeks of monitored abstinence. Notably, more severe cannabis use markers (greater lifetime use, CUD symptoms, and shorter abstinence) were linked with this reduced connectivity. Thus, findings support public policy aimed at reducing and delaying cannabis use and treatments to assist with sustained abstinence. Future longitudinal studies are needed to investigate causation.

## 1. Introduction

Cannabis is the second most widely used drug in the United States by adolescents and young adults with a lifetime prevalence rate of 15.4% for adolescents and 51.5% for young adults [1,2]. As adolescence is a time of critical brain development [3], specifically within the endocannabinoid (eCB) system [4,5,6,7], adolescents may be significantly impacted by early, repeated cannabis use. Despite these concerns, neurocognitive alterations linked with cannabis exposure following monitored abstinence in adolescents is understudied and findings to date are inconsistent. 

The primary psychoactive component of cannabis is Δ^9^-tetrahydrocannabinol (THC) that activates the endogenous endocannabinoid (ECB) system. THC interacts and binds to the cannabinoid receptor 1 (CB1) and animal research has demonstrated that repeated cannabis results in downregulation or desensitization of the CB1 receptor [8]. Notably, the ECB system continues to develop throughout adolescence into young adulthood [3,9]. Repeated use during adolescence may result in alterations in regions rich in eCB signaling, including the parietal cortex [10,11,12], a region that continues to develop into adolescence and underlies sustained attention [13]. Consistent with this, several neuropsychological studies have shown evidence that young regular cannabis users, typically defined as at least weekly cannabis use, have demonstrated poorer sustained and selective attention, attentional control, and increased lapses in attention [14,15,16,17,18,19,20,21,22,23,24,25,26,27,28,29,30,31,32,33,34,35,36]. Thus, repeated exposure to cannabis during the neurodevelopmental phase of adolescence may result in alterations in the attentional network. 

One criticism of the current literature is that cannabis effects may only be observed within a few days of use, and this may be related to either very acute effects or even cannabis withdrawal versus more chronic impact [37]. Indeed, some research has suggested, in humans, there is evidence of recovery in CB1 receptor density even only after a few days of cannabis abstinence [38]. In addition, other research demonstrates recovery of cognitive function, particularly on memory-based tasks [39]. For example, Schweinsburg et al., (2010), [40] found that among heavy adolescent cannabis users who had remained abstinent (between 27 and 60 days) showed improvements in a spatial working memory task relative to recent cannabis users (between 2 and 7 days). Some studies have found attentional abnormalities measured on neuropsychological tests in cannabis using adolescent and young adults compared to controls after two to three weeks of monitored abstinence [22,28,33,41]. Although [33] Hanson et al. (2010) found improved verbal working memory after a three-week period of abstinence among adolescent cannabis users (ages 15–19), however, attention accuracy on a sustained and selective attention task remained impaired. Notably, our group has found improvements in a sustained attention task after two weeks of abstinence among a slightly older sample of adolescent and young adult cannabis users (ages 16–26), but found no significant difference between controls and cannabis users on verbal working memory and verbal learning tasks [16]. However, at a later time point in this same sample, our group also found evidence of deficits on another sustained attention task (CPT) [41], as well as working memory and psychomotor speed [23]. Thus, some evidence suggests attentional differences remain even after sufficient time for THC metabolites to be eliminated from the body and withdrawal symptoms have generally recovered [42,43]. Taken together, previous evidence suggests that impairments in sustained attention among cannabis users may take longer to recover compared to deficits shown in verbal or working memory. Despite these findings, few studies to date have examined underlying neuronal signaling in the attentional networks following a period of monitored abstinence. 

It is theorized that there are two anatomically and functionally segregated but complementary attention networks [44], the dorsal and ventral attention networks (DAN and VAN). The DAN employs dorsal fronto-parietal areas which involves top–down visual attentional control and voluntary allocation of attention. The visual attention system is a top–down system controlled by both cognitive factors (i.e., prior knowledge, expectation, and current goals), and bottom–up factors reflecting sensory stimulation [44]. The DAN consists of the bilateral inferior parietal sulcus (IPS) and bilateral frontal eye fields (FEF) [44]. The DAN is activated when there is prior knowledge or expectation of seeing an object in a particular location, for example, movement in a certain direction [45]. The DAN is also involved in associating relevant stimuli to responses and is activated during task preparation [46]. The DAN responds alongside the VAN, a secondary system including the ventral fronto-parietal areas. The VAN is activated during this bottom–up processing of new or unexpected sensory stimuli [44]. This processing of new and behaviorally significant stimuli can cause a shift in attention or goal directed processes [44]. The VAN is theorized to be involved when an important stimulus appears outside the context of the current task or focus of attention, and the VAN initiates a reorienting or “circuit-breaking” to interrupt the dorsal signaling, ultimately redirecting attention [47]. Structurally, the VAN is localized in the right hemisphere and consists of the right temporoparietal junction (TPJ) and the right ventral front cortex (VFC) [44,45]. 

The developmental trajectory of both attention networks continues through to young adulthood [48]. Resting state functional connectivity studies have found an inverted U-shaped trajectory of the DAN through development, as the DAN connections become more dominant throughout childhood, until the age of 30 in which synaptic pruning eliminates many short-range connections from development [49,50]. Specifically, there is stronger within-network connectivity in the DAN among adolescents aged 11–13 relative to 19–25-year-olds [51]. Particularly, for the DAN, relative to adolescents, adults show greater functional connectivity between the FEF and the posterior cingulate cortex [52]. As the DAN becomes more dominant through childhood and eventually pruned in adulthood, the VAN shows greater functional connectivity among adults relative to children [52,53]. For the VAN, relative to adults, adolescents show greatest functional connectivity among the VFC and nodes related to salience, including the anterior insula and the anterior cingulate cortex (ACC), which may demonstrate a reduced segregation of these networks in childhood relative to adulthood [52]. Given this ongoing development, exposure to exogenous cannabis may alter this development of attentional networks in regular cannabis using youth.

To better understand these attentional networks, resting state functional connectivity (RSFC) can be used to measure the synchrony of regional brain activation during rest, and are particularly useful in the absence of a task [54,55,56,57,58,59]. In previous studies, the DAN and VAN are shown to be consistently segregated systems using RSFC [60,61]. Previous research has found that the DAN and VAN are associated with performance on behavioral tasks including the Attention Network Test-Revised test, a task assessing ability of alerting, orienting, and other attentional functions [62] among healthy adults. Functional MRI studies have shown associations of the connectivity of the DAN and VAN associated with behavioral performance on the Attention Network Task, suggesting that the DAN and VAN are correlated with alerting, orienting, and executive control among healthy controls [63], indicating an intrinsic functional network and important neuronal mechanism in attention. Additionally, other work has shown associations with the DAN and VAN with other sustained attention tasks including the gradual Conners’ Continuous Performance Test (grad CPT) and the Test of Variables of Attention (TOVA) [64]. Other studies have shown intraindividual fluctuations in accuracy on attention tasks associated with the DAN [65,66,67] that may be explained by low DAN connectivity can precede an attention lapse. Additionally, a previous study demonstrated that functional connectivity in the DAN is associated with selective attention skills in early childhood [68]. A study assessing functional connectivity analyses among the DAN and VAN networks and the default mode network (DMN) found a strong negative relationship between the DMN and both attentional networks, and this negative relationship was found to have a robust relationship with behavioral performance on the flanker task [69]. Specifically, the strength of this negative correlation varied among individuals’ response times of the flanker task and were related to intraindividual variability, which can be a useful index in how efficient attention is regulated, particularly as an indicator of neurological dysfunction [69]. 

Although there is strong behavioral evidence to support alterations in attentional tasks and performance among cannabis users, there have been limited studies using resting state functional connectivity (RSFC) to measure the DAN and the VAN connectivity among adolescent and young adult regular cannabis users. In a study of, primarily, male young adults (*n* = 103, 18–38 years old) with and without both early psychosis and cannabis use history, controls with a history of cannabis use showed increased connectivity in the DAN compared to non-using participants, and DAN strength positively correlated with severity of cannabis use dependence [70]. Notably, no abstinence period was required, the sample included participants outside of the young adult developmental period and was primarily male; thus, findings may not generalize to younger samples. [71] Peeters and colleagues (2015) found that cannabis use did not moderate the relationship between psychosis risk and functional connectivity in frontoparietal regions [71]; they did not find any association or moderation of cannabis use. Notably, this study was limited to the frontoparietal region and did not specifically distinguish the DAN or VAN networks. Thus, due to the paucity of this research, further examination of RSFC among DAN and VAN and how differences in connectivity may correlate with cannabis use symptoms may shed light in determining mechanisms of cannabis use effects on attentional neural networks.

The primary goal of the present study was to investigate the effects of regular, or weekly, cannabis use on the dorsal and ventral attentional networks using seed-based resting-state functional connectivity in adolescents and young adults after a two-week period of monitored abstinence. The secondary goal explored whether the corresponding differences are associated with cannabis use patterns and behavior. We hypothesized that increases in connectivity within the dorsal (DAN) and ventral (VAN) attentional networks will be demonstrated for chronic cannabis users. Additionally, we hypothesized that the increase in intraconnectivity within either the DAN or VAN will be associated with increased cannabis use severity, decreased length of abstinence, earlier age of regular use onset, and increased severity of Cannabis Use Disorder (CUD) symptoms. 

## 2. Participants and Methods

### 2.1. Participants 

Cannabis-using and non-using participants were recruited for a larger parent study through media advertisements, flyers posted around universities, cafés, bars, headshops, recreation centers, and festivals. A total of 75 participants (36 cannabis users and 39 non-using controls; 53.3% males; aged 16–26 years of age) participated in the current study. Racial identities consisted of the following: predominantly Caucasian (66.7%), Asian (12.0%), Multi-racial (8.0%), African American (8.0%), and American Indian/Alaska Native (1.3%). In addition, 14.6% of participants identify as Hispanic/Latino, 84% are not Hispanic/Latino, and 1.3% are Unknown. Cannabis-users were defined as using cannabis at minimum 50 times in the past year, or 100 times in their lifetime. Non-using controls were defined as using cannabis less than five times in the past year, or less than 20 times in their lifetime. Given that there may be minor cognitive and structural changes of one or two instances of cannabis use among youth [72], controls who had used in the past 30 days according to the Timeline Followback and had any positive toxicology screenings via sweat patch or urine at session one were excluded.

Participants were included in the study if they were right-handed, spoke English, and were willing to abstain from substance use for three weeks. Exclusionary criteria included MRI contraindications (pregnancy, claustrophobia, or metal in the body), reported prenatal alcohol (>6 drinks per week or >4 per day) or nicotine exposure, birth complications, premature birth (<33 weeks gestation), significant prenatal health problems (gestation < 35 weeks), major neurological disorders (e.g., seizures, migraines, tumors, chemotherapy, multiple sclerosis, movement disorders), loss of consciousness > 2 min, history of a learning disability or intellectual disability, major health problems (e.g., metabolic disorders), independent non-substance induced DSM-IV Axis I disorder diagnosis (aside from a substance use disorder), current use of psychoactive medication, and heavy other drug use (>20 lifetime uses of each non-cannabis drugs). 

### 2.2. Procedures 

Data were collected from a larger parent study examining the effects of physical activity and cannabis use on neurocognitive outcomes among adolescents and young adults (R01 DA030354; PI: Lisdahl). The study received approval by local institutional review board. All participants and parents or participants under the age of 18 participating in the study provided informed written consent/assent. In order to assess inclusion and exclusion criteria, both the youth participant and a parent/guardian (required for ages 16–17; highly preferable for ages 18–25) were screened by trained research assistants. Eligible participants completed a total of five separate sessions over the course of four weeks. Sessions 1–3 were weekly sessions consisting of drug testing (sweat patch continuously plus urine toxicology testing at sessions) and brief psychological and neuropsychological testing. Sessions 4 and 5 were conducted one week after Session 3 and within 24–48 h of each other. Session 4 included a 3 h neuropsychological battery, while Session 5 consisted of MRI scanning (both also included urine toxicology testing to cover the 24–48 h between sessions). Those who newly tested positive for drugs and/or alcohol (excluding nicotine) during Sessions 2–5 were allowed to continue participation in the study if over the course of the study, THC levels continued to drop via PharmChek testing. If positive testing occurred for any other substance, THC levels increased, or breathalyzer concentration was higher than 0.000, the participant was excluded from the analysis. All participants underwent a minimum of two weeks of monitored abstinence prior to the neuropsychological and MRI sessions. Subjects were compensated for their involvement in the study.

### 2.3. Measures

#### 2.3.1. Detailed Phone Screen

Lifetime Substance Use Patterns. Overall patterns of substance use were assessed using the Customary Drinking and Drug Use Record (CDDR) at the phone screening to measure quantity/frequency of cannabis, nicotine, alcohol and other drugs, age of regular use of cannabis (once per week), withdrawal symptoms, DSM-IV CUD criteria symptoms counts, and substance-use related difficulties [73,74]. 

Mini Psychiatric Interview. Semi-structured interviews were conducted by trained research assistants to diagnose Axis I disorders were administered using the Mini International Psychiatric Interview (MINI) [75] for participants 18 years or older, and individuals between ages 16 and 17 were given the Mini International Psychiatric Interview for Children and Adolescents (MINI-KID).

#### 2.3.2. Study Session 

Demographic Information. Participants were given a background questionnaire [76], including demographic variables such as age, gender, race/ethnicity, self and biological parents’ educations, incomes and employments, marital status, history of medical or neurological illness, psychological disorders or use of psychiatric medication, and learning disability. 

Substance Use Toxicology Measurement. Participants were expected to remain abstinent from alcohol and other drugs (except tobacco) throughout the course of the study, thus abstinence was evaluated at each session through a variety of measures. A PharmChek drug patch was worn by all participants throughout the duration of the study to ensure abstinence via sweat from substances that would not be gathered through the weekly urine screening. The Pharmchek drug patches were used to detect cocaine, benzoylecgonine, heroin, 6-monoacetylmorphine, morphine, codeine, amphetamines, methamphetamine, THC, and phencyclidine (PCP) at a minimum cut off of 0.5 ng/mL for THC, 7.5 ng/mL for PCP, and 10 ng/ML for all other drugs tested. At each session, urine samples were used to measure cotinine (a metabolite of nicotine) levels with NicAlert test strips, and the ACCUTEST SplitCup 10 panel drug test screened for amphetamines, barbiturates, benzodiazepines, cocaine, methamphetamines, PCP, ecstasy, methadone, opiates, and THC carboxylic acid (THC-COOH) through the same urine sample provided. To test for recent alcohol consumption, all participants completed a breathalyzer test at every session.

Past Year and Lifetime Substance Use. An adjusted Timeline Followback (TLFB) [77,78] interview was conducted by trained research assistants to measure substance use patterns on a week-by-week basis over the past year. Substance use was measured in standard units including cannabis (number of joints or milligrams in concentrates; concentrate use was then converted to joins for overall cannabis measure), alcohol (standard drinks), nicotine (number of cigarettes, hits of chewing tobacco, snuff, cigars, pipes, or hookah), ecstasy (number of tablets), sedatives (number of pills or hits of GHB), stimulants (cocaine, crack cocaine, and methamphetamine converted into milligrams), hallucinogens (number of hits or occasions of PCP, LSD, DMT, peyote, and salvia), opioids (number of hits of opium or heroin), and inhalants (number of hits). Length of cannabis abstinence was calculated as days from date of last cannabis use based on the TLFB and date of scan. 

### 2.4. MRI Scan Acquisition and Pre-Processing

MRI Scan Acquisition. Structural MRI scans were administered with a 3 T Signa LX MRI scanner (GE Healthcare, Waukesha, WI) using a 32-channel quadrature transmit/receive head coil. High-resolution anatomical images were acquired using a T1-weighted spoiled gradient-recalled at steady-state (SPGR) pulse sequence (TR = 8.2 ms, TE = 3.4 s, TI = 450 and flip angle of 12°). The anatomical images had an in-plane resolution of 256 × 256 with a square field of view (FOV) of 240 mm. A total of 150 slices were acquired at 1 mm thickness. Functional resting-state MRI scans were acquired with gradient-echo echo planar imaging (EPI) pulse sequences in the sagittal orientation (TR = 2 s, TE = 25 ms and flip angle = 90°). The in-plane resolution was 64 × 64 with a FOV of 240 mm with 40 contiguous 3.7 mm slices. Participants were told to lie horizontally with their eyes closed while remaining awake for an eight-minute resting scan.

MRI Preprocessing. fMRI data were processed using Analysis of Functional NeuroImages (AFNI) software [79]. The Configurable Pipeline for Analysis of Connectomes (https://fcp-indi-github.io, accessed 16 February 2022) was used as the automated pipeline for motion correction, nonparametric non-uniform intensity normalization, Montreal Neurologic Institute (MNI152) transformation, removal of non-brain materials, skull-stripping, and topology correction. The first three time points were removed from each resting-state scan due to T1 stabilization effects. Each subject’s resting state-MRI scan was aligned to the mean intensity values over all time points for each voxel to motion correct within the scan [80]. Linear and quadratic detrending was applied to remove residual drift. All functional data were smoothed in-plane using a 6-mm full width half maximum Gaussian kernel and then temporally filtered (0.005 < f < 0.1). The resulting registration matrix was applied to the resting-state scan. Nuisance signals, including white matter and CSF signals, and six motion parameters were regressed from the data [59,81].

### 2.5. Data Analysis

ANOVAS and chi-squares were run to test differences between cannabis users and controls.

Primary Aim. RSFC MRI Analysis. To examine the functional connectivity within the attention networks, whole brain seed-based resting state functional connectivity analyses were performed using seeds within the DAN and VAN (see Figure 1). The 6-mm radius spheres were used as seeds, centered on the right inferior parietal sulcus (IPS; 27, −58, 49) and the frontal eye fields (FEF; 24, −13, 51) [60], both within the DAN. Seeds were also centered on the right ventral frontal cortex (VFC; 37, 18, 1) and the right temporal-parietal junction (TPJ; 53, −48, 20) [60] both within the VAN. The correlation between the mean time series within the seed centered on IPS, FEF, VFC, or TPJ and the entire cortex was the basis of the seed-based RSFC. The correlation coefficients were transformed to Fisher Z-scores using AFNI software [79]. Changes in functional connectivity within the DAN and VAN were examined by comparing the connectivity within each attentional networks between the control group and cannabis using group, controlling for past year alcohol and nicotine cotinine levels. A family-wise error (FWE) threshold of pFWE < 0.05, (individual voxel threshold at *p* < 0.005) was applied using a cluster-threshold method of correcting for multiple comparisons using Monte Carlo simulations within 3dClustSim [82], with individual voxels labeled significant at *p* < 0.005, corrected for Family-Wise Error (FWE) at cluster thresholds of *p* < 0.05. This methodology for cluster thresholding has been shown to effectively control false-positive rates [83,84]. 

Secondary Aim. Brain-Behavior Correlation Analysis. Regions that were significantly different between cannabis users and controls were extracted and used for a correlation analysis to examine brain–behavior relationships in dose-dependent fashion across all participants. IBM SPSS Statistics for Windows, version 28.0 was used to calculate the Pearson’s correlation coefficient between the mean Z-score of each significant RSFC regions past year and lifetime cannabis use, length of abstinence, age of regular use onset, and Cannabis Use Disorder (CUD) symptoms.

## 3. Results

### 3.1. Demographic and Substance Use Characteristics

There were no significant differences observed between the two groups for age [F(1, 74) = 0.287, *p* = 0.594], race (*χ*^2^ (1) = 6.944, *p* = 0.326), ethnicity [(*χ*^2^ (1) = 2.098, *p* = 0.350), gender (*χ*^2^ (1) = 3.099, *p* = 0.078), or years of attained education [F(1, 74) = 1.512, *p* = 0.223] (see Table 1). 

Substance Use. Cannabis using individuals consumed more cannabis [F(1, 74) = 35.232, *p* < 0.0001], alcohol [F(1, 74) = 17.628, *p* < 0.0001], and cigarettes [F(1, 74) = 7.569, *p* = 0.007], over the previous year than controls. Additionally, cannabis using individuals had higher cotinine levels the day of the MRI scan [F(1, 74) = 6.518, *p* = 0.013]. Therefore, past year alcohol use and cotinine levels on the day of the MRI scan were used as covariates in all analyses.

### 3.2. RSFC Differences between Cannabis Users and Controls

#### 3.2.1. RSFC Differences within DAN

Seed: IPS. After controlling for past year alcohol use and cotinine levels, cannabis users had increased negative RSFC between the IPS seed and right anterior insula and overlapping part of white matter compared to controls (MNI Coordinates: 27, 24, 12) (F = 12.897, *p* < 0.05), indicating that cannabis users had lower functional connectivity within the DAN (see Figure 2). This activation touches part of the anterior insula, however the majority of the cluster is in the white matter.

Seed: FEF. There were no significant differences among cannabis and controls in functional connectivity in the FEF seed.

#### 3.2.2. RSFC Differences within the VAN

Seed: VFC: There were no significant differences among cannabis and controls in functional connectivity in the TPJ seed. 

Seed: TPJ. There were no significant differences among cannabis and controls in functional connectivity in the TPJ seed. 

#### 3.3.3. Correlations between RSFC Networks (DAN) and Cannabis Use Patterns

Increased negative connectivity coefficient between the right IPS and right anterior insula and white matter region was correlated with increased lifetime cannabis use (r = −0.318, *p* = 0.001) and lifetime Cannabis Use Disorder Symptoms (r = −0.346, *p* = 0.05) and shorter length of abstinence (r = 0.306, *p* < 0.001) (see Figure 3) in cannabis users and non-using controls (*n* = 49). 

## 4. Discussion

This study aimed to investigate dorsal (DAN) or ventral (VAN) attentional network RSFC in regular adolescent and young adult cannabis users after they achieved sustained, monitored abstinence for a two-week period. Further, we investigated relationships with dose-dependent cannabis use patterns and symptoms. We found lower functional connectivity within the DAN network in cannabis users compared to controls. This increase in negative connectivity in the DAN was linked with heavier lifetime use patterns, greater Cannabis Use Disorder (CUD) symptoms, and shorter length of cannabis abstinence.

Our results demonstrate subtle aberrant and reduced connectivity in the DAN network, specifically between the right IPS and parts of the right anterior insula and adjacent white matter in cannabis users relative to non-cannabis users, even after at least two weeks of monitored abstinence. The dorsal attentional network is posited to regulate endogenous signals that are related to top-down modulatory signals biasing attention and processing towards a particular stimuli [85,86]. The right anterior insula has been posited as central components in cognitive control, specifically decision making and executive functioning [87]. Furthermore, alterations of the insula has been implicated previously in cannabis users [88,89] and plays a key role in the development and underlying aspects of substance use disorders [90]. The region also included a small portion of the anterior cingulate (ACC); the anterior insula’s connectivity has major bidirectional connections with the anterior cingulate cortex (ACC) [89]. Previous work in our lab has shown increased intrinsic bilateral left and right ACC connectivity [91], abnormal ACC connectivity in response to cognitive inhibition fMRI task [16], and reduced left and right rACC volume among young cannabis users [92]. [72] Basseer Sami and colleagues (2020) found increased connectivity among cannabis users in the DAN relative to non-cannabis users, although there were several notable differences between the study samples including inclusion of psychosis and older age. These findings also build upon previous work demonstrating differences in behavioral measures of attention in chronic cannabis users compared to controls [28,32,33,93]. Therefore, current results extend upon previous findings to show that repeated cannabis use may disrupt the DAN network, potentially playing a role in one’s ability to focus their attention on an object and on unexpected or unattended sensory stimuli. These findings are particularly important as there is evidence of comorbidity between cannabis use and Attention Deficit Hyperactivity Disorder (ADHD) [94], despite links between cannabis and poorer executive functioning and attention in those with ADHD diagnosis and symptomology [41,95]. Thus, it is recommended that clinicians, pediatricians, and health professionals that engage with youth provide prevention education to youth and caregivers regarding the relationships between cannabis use and potential attentional disruptions. 

The current study also examined the influence of more detailed cannabis-related variables in DAN regions that differed according to group. We found significant post hoc relationships between decreased functional connectivity of the DAN (right IPS with right anterior insula) and increased lifetime cannabis use exposure, CUD symptoms, and shorter length of abstinence. Most notably, our findings are consistent with prior studies demonstrating functional connectivity of the DAN is associated with behavioral aspects of cannabis use severity and symptomatology [70] and may indicate a dose-dependent relationship between cannabis use exposure and aberrant connectivity in the DAN. Further, disrupted connectivity in the DAN and the right anterior insula may be particularly important to investigate as a mechanism for increased risk of CUD symptoms among youth. Notably, these relationships were only examined in the regions that significantly differed by group in order to aid interpretation; additional large-sample, whole-brain analyses need to be conducted to further examine dose-dependent effects. Finally, greater length of cannabis abstinence (31–313 days) was associated with increased or more normalized DAN functional connectivity, providing hopeful evidence for recovery of function following sustained cannabis abstinence. Although previously studies found that cannabis users experienced some recovery of sustained attention abilities during a two-week abstinence period [16,32,96], other domains of attention remained abnormal following one to three weeks of abstinence [27,28,41]. These correlational findings reveal some preliminary evidence for potential functional connectivity recovery. More work specifically utilizing repeated brain imaging to investigate whether brain structure and function recover with monitored cannabis abstinence are needed. 

There are several potential mechanisms underlying abnormal connectivity between the right IPS and right anterior insula among young cannabis users. Prior preclinical research has suggested cannabis use during adolescence and young adult years leads to alterations of the CB1 receptor density [97] and evidence has shown several cannabinoids interfere with 2-AG levels and CB1 expression in animal models, specifically in the insula [5,98]. Repeated cannabis use may also impact gamma-Aminobutyric acid (GABA) and glucose transporter (GLUT) signaling, including in the ACC [99]. CB1 density, eCB signaling, and GABA/GLUT disruption may ultimately impact the timing of neuronal signaling or alter underlying gray matter and white matter structure within inferior frontal gyrus and parietal cortex. Prospective, longitudinal studies conducted prior to the onset of cannabis initiation that further investigate these mechanisms are needed to determine causation and clarify underlying mechanisms.

Some limitations of the current study should be noted. First, due to the cross-sectional nature of the study, we cannot determine timing or causality of cannabis use and RSFC of the DAN and VAN. Thus, the abnormalities found to the RSFC of the DAN could possibly be related to risk for initiating or sustaining cannabis use. Prospective, longitudinal studies conducted prior to the onset of cannabis use, such as the Adolescent Brain Cognitive Development (ABCD) Study (https://abcdstudy.org/, last accessed 16 February 2022), are needed to disentangle causality [100]. Secondly, while these results adhered to a *p*-value of 0.005, these findings were no longer significant after thresholding at a *p*-value of 0.001. Thirdly, this region includes the right anterior insula, however also includes some adjacent white matter and encroaching into the anterior cingulate cortex. Thus, this may be a limitation to interpreting these findings and future studies are needed to replicate these findings at a threshold of 0.001 and with a larger sample size. Additionally, more detailed measures of cannabis exposure were only examined in regions that differed according to group in a post hoc analysis; additional large-scale studies are needed to tease apart unique contributions of cannabis use (e.g., dose, potency, mode of use, CUD symptoms) on DAN connectivity patterns. This study investigated cannabis use behaviors associated with significant group resting state functional connectivity in the right anterior insula, which may increase the likelihood of finding significant brain–behavior relationships across groups which may pose as a limitation of this study. Furthermore, these findings should not be taken as evidence that these relationships solely exist in these significant regions, but to aid in the development and interpretation for future research. Future studies are needed that include the investigation of within-subjects to investigate whether consistent use of cannabis use is related to reduced connectivity in this region. Finally, this study is limited to a minimum of two weeks of abstinence. This time period ensures that the abnormalities seen in attention were not due to acute withdrawal effects [43], but findings may not generalize to participants with shorter or longer lengths of abstinence. Finally, cannabis users had comorbid alcohol and nicotine use; although these were statistically controlled for in the analyses, results should still be interpreted within the context of potential polysubstance use.

## 5. Conclusions

In conclusion, the current study provides additional evidence that regular cannabis use during adolescence and young adulthood is associated with subtle, abnormal connectivity within the DAN attentional network after at least two weeks of monitored abstinence. Further, youth who used more cannabis, had more symptoms of CUD and shorter lengths of abstinence demonstrated the greatest abnormalities. Therefore, public health campaigns should continue to focus on reducing the impact of heavy cannabis use in youth and encourage abstinence. Future research should investigate the mechanisms in which cannabis use patterns contribute to or predict differences in attentional neural networks in a prospective, longitudinal design.

## Figures and Tables

**Figure 1 brainsci-12-00287-f001:**
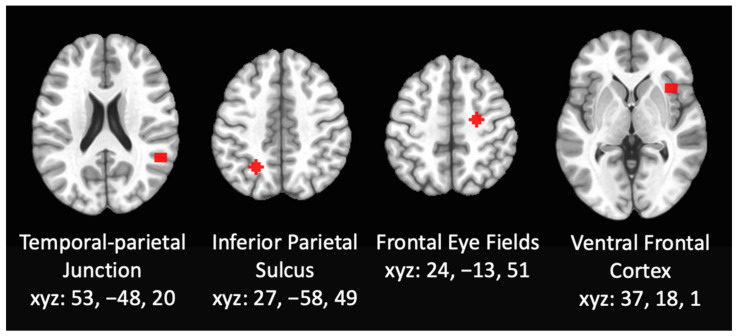
Resting state functional connectivity was assessed using whole brain seed-based analyses using seeds within the DAN and VAN including 6-mm radius spheres centered on the right inferior parietal sulcus, the frontal eye fields, the right ventral frontal cortex, and the right temporal-parietal junction.

**Figure 2 brainsci-12-00287-f002:**
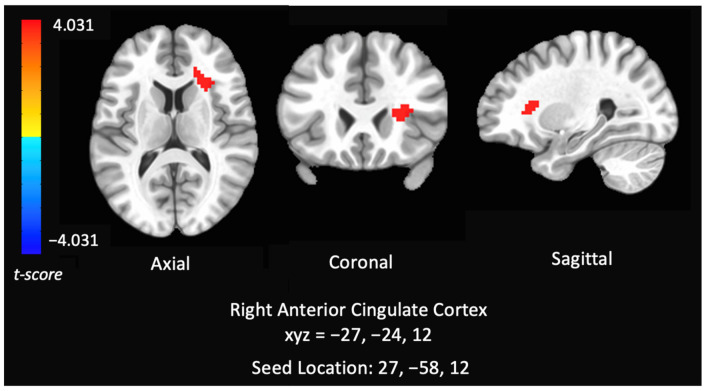
Group analysis of connectivity within the dorsal attention network of controls versus chronic cannabis users. Comparison of whole-brain resting state functional connectivity of controls versus chronic cannabis users with seed regions within right inferior parietal sulcus. The colors represent areas of significant connectivity; warm colors indicate increased connectivity in controls compared to cannabis users. The axial, coronal, and sagittal views are featured (left = left).

**Figure 3 brainsci-12-00287-f003:**
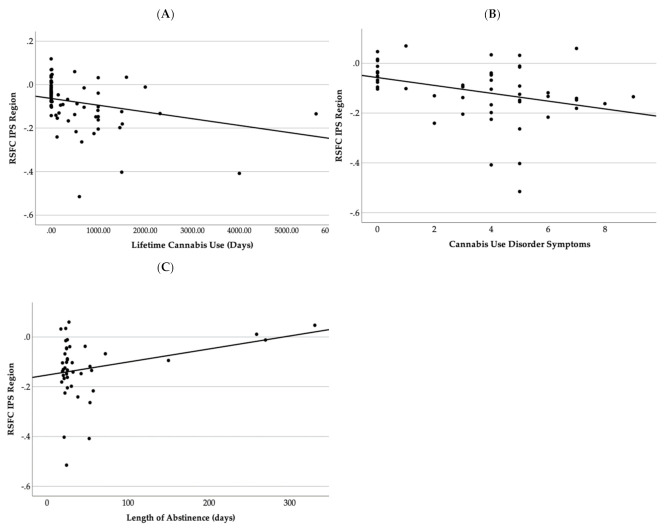
This figure demonstrates a scatter plot graph between resting state functional connectivity Fisher Z-scores and (**A**) lifetime cannabis use, (**B**) Cannabis Use Disorder symptoms, and (**C**) length of abstinence in days.

**Table 1 brainsci-12-00287-t001:** Demographic and drug use characteristics.

	Controls	Cannabis Users
	(*n* = 39)	(*n* = 36)
Age M (SD)	20.95 (2.67)	21.25 (2.16)
Education M (SD)	14.31 (2.32)	13.75 (1.48)
Gender (% Female)		
Male *n* (%)	17 (43.6)	23 (63.9)
Female *n* (%)	22 (56.4)	13 (36.1)
Race (%)		
American Indian/Alaska Native	0 (0.0)	1 (2.8)
Asian	6 (15.3)	3 (8.4)
Native Hawaiian/Other Pacific Islander	1 (2.6)	0 (0)
Black or AA	2 (5.1)	4 (11.1)
White, Caucasian, not of Hispanic Origin	28 (71.8)	22 (61.1)
More than on race	1 (2.6)	5 (13.8)
Unknown	1 (2.6)	1 (2.8)
Ethnicity %		
Hispanic/Latino	4 (10.2)	7 (19.4)
Not Hispanic	34 (87.2)	29 (80.6)
Unknown	1 (2.6)	0 (0.0)
Past Year Cannabis Use (joints) M (SD)	0.40 (1.16)	421.69 (443.50)
Length of cannabis abstinence (days) M (SD)	168.33 (132.57)	32.06 (23.24)
Minimum	31	17
Maximum	313	150
Past Year Alcohol Use (standard drinks) M (SD)	93.65 (143.59)	315.36 (294.10)
Past Year Cigarette Use (cigarettes) M (SD)	0.55 (2.00)	213.59 (484.08)

## Data Availability

Requests for data are welcomed and can be submitted to the corresponding author.

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
