# Peer review of "Disrupted Resting State Attentional Network Connectivity in Adolescent and Young Adult Cannabis Users following Two-Weeks of Monitored Abstinence"

_brainsci, 2022, doi:10.3390/brainsci12020287_

Round 1

Reviewer 1 Report

In this paper, the authors used seed-based resting state functional connectivity to examine differences in attention networks between cannabis-using young adults and controls. The observed differences in the dorsal attention that they claim relate to measures of use. In general, this is an important topic and the paper is written well enough. However, there are a couple of major issues that prevent me from recommending publication.

Major issues

  1. The authors claim there is a relationship between rsFC in DAN and cannabis use measures. This analysis is in the methods, and the result is discussed. However, this result is missing from the results section altogether. It is not described there, nor are there any figure(s) supporting that result.
  2. The authors claim the group difference lies in the right anterior cingulate cortex. However, the c-ordinates (x = -27) and figure show this far off the midline that would be indicative of ACC. In the slice presented, the region lies nearly completely in white matter, at best touching insula. Further, the software they said the used (AFNI), puts the center of the region closest to opercular regions.
  3. The authors did not correct for testing 4 different seed locations. That, coupled with the relatively anticonservative voxel-wise threshold (we note that the Cox 2017 paper they cite recommend voxel-wise p=.001 or p=.002) and the results being largely in white matter makes it difficult to have confidence in these results.

Minor issues

  1. Line 174: The authors claim .01% participants have unknown ethnicity. This is not a possible percentage with 75 participants
  2. Line 176: The choice of controls seems strange. As the authors say in the intro, about 50% of young adults have used cannabis – leaving about 50% w/o use. Although controversial, Orr et al 2019 showed brain alterations associated with minor lifetime usage. So, the up to 20 lifetime uses in the present control cohort could still be consequential.
  3. Line 304: I believe IBM owns SPSS, not Microsoft
  4. The authors frequently use relative language when describing differences (e.g decreases, reduces). This language implies a pre-use baseline (i.e. it implies that the use caused the change). Given the lack of that baseline, the authors should use absolute language in these cases (e.g. “the cannabis users had lower rsFC…”)
  5. The authors say the main results is at MNI -27, -24, 12. These results appear to be in RAI orientation. Most people would assume MNI coordinates are in LPI orientation (i.e. the authors should flip the first 2 signs).
  6. Figure 1 would be improved with an inset showing the seed location, a left/right indicator and a colorbar (although the results appear to be one color, the legend indicates that a range is possible).

Reviewer 2 Report

Review of manuscript –

Disrupted resting state attentional network connectivity in adolescent and young adult cannabis users following two-weeks of monitored abstinence

This manuscript presents the need and outlines the findings, to a study of functional connectivity during resting state of two attentional networks in adolescent and young adult cannabis users. The paper as a whole is well-thought-out, clear in its intention and purpose, and gives a commendable introduction and context to the topic. Where this paper is lacking is in the lack of presentation regarding results from analysis looking at cannabis use parameters in correlation with functional connectivity findings.

  • The manuscript would benefit from a figure to display the four seeds being used for both DAN and VAN analysis
  • A small amount of revision required to the text in order to make sure it read coherently – the first line of the discussion for example.
  • The secondary aim of the paper indicates exploration of the relationship between cannabis use parameters (life-time use, ago of onset) – these results are not presented in the results. Was this analysis carried out? There is a reference to them in the discussion. Description of the results from this analysis in the results section supported by a table would be a valuable inclusion to the manuscript.
  • In line 386, an analysis of the effects of abstinence is introduced. Is this finding from the analysis being presented in this paper? This is an important finding and should be presented in the results.

Round 2

Reviewer 1 Report

While the authors addressed many of my concerns, not all were adequately addressed. The numbers below refer to my original review

  1. The added results are interesting, but really should be accompanied with a figure showing the regressions. A figure contains much more information that the simple Rs reported. Further, table 2 should be removed as it adds nothing (the 3 numbers in it are in the preceding paragraph). Finally, there is a mismatch in the results table and paragraph. The former shows a correlation between CUD symptoms as Lifetime Use as 0.306 where the latter has the correlation between rsFC and length of abstinence as 0.306.
  2. As a point of clarification, I did not mean to imply that the activation is in the insula. In the slice presented, it looks almost entirely in white matter. It may be that presenting a montage of slices will help the reader better understand where exactly the activation lies. As presented, given the white matter overlap and the lack of any result at a more conservative threshold, I’m inclined to think it is likely that the presented region is a false positive.
  3. I assume that 1 of the 75 participants have an unknown race. That would be 1.3%, not .013%.
  4. It has been years since I used SPSS, but I recall (and Wikipedia agrees) that the version number is in the 20s, not version 2 that you state.

9. The colorbar (too big, in my opinion) should have numbers at the ends to indicate the meanings of the colors.

Round 3

Reviewer 1 Report

The author’s changes have improved the paper. However, the inclusion of figure 2 has revealed an inconsistency and, I believe, a methodological error. On line 307 you state “…examine brain-behavior relationships within each group” (my emphasis). However, the results in the figure include data points for all participants (i.e. both groups pooled). This is problematic as it is a circular analysis. You picked those regions because they differed in activation between the groups. As the behaviors differed between the groups, these regions will be (strongly) biased toward having a significant regression.

In other words, you first did (written in regression notion, but a t-test is equivalent to a regression with a dummy-coded variable)

brain ~ group

and then in regions that were (highly) significant, you did

brain ~ behavior

on behavior that differed between groups. That variable essentially codes for group, biasing you to finding a significant result.

The analysis you did only makes sense if you do it only in the ‘cannabis users’ group, like you imply in the methods. Note doing it in the controls groups separately does not make sense as they lack variability in the independent variable. Again, doing it in a combined group is statistically problematic.

Additionally, it is completely unclear to me which data went into the ‘Length of Abstinence’ regression as I count at least 40 data points, and maybe a few more overlapping. So, not both groups combined but also not just the users.

Finally, some minor notes about this figure. In your results (lines 347-51), you say “decreased connectivity…” In reality, the relationships are consistent with an increase in negative connectivity. Also, I don’t know what is meant by ‘a proper scatter plot graph’ in the caption; this should be removed (assuming the results hold within users only).
